# Antioxidant, Antibacterial Activities and Mineral Content of Buffalo Yoghurt Fortified with Fenugreek and *Moringa oleifera* Seed Flours

**DOI:** 10.3390/foods9091157

**Published:** 2020-08-21

**Authors:** Faten Dhawi, Hossam S. El-Beltagi, Esmat Aly, Ahmed M. Hamed

**Affiliations:** 1Agricultural Biotechnology Department, College of Agriculture and Food Sciences, King Faisal University, P.O. Box 420, Al-Ahsa 31982, Saudi Arabia; falmuhanna@kfu.edu.sa; 2Biochemistry Department, Faculty of Agriculture, Cairo University, Giza 12613, Egypt; 3Dairy Technology Research Department, Food Technology Research Institute, Agricultural Research Center, Giza 12613, Egypt; esmat_rayan_2010@yahoo.com; 4Dairy Science Department, Faculty of Agriculture, Cairo University, Giza 12613, Egypt

**Keywords:** functional yogurt, fenugreek and *Moringa oleifera* seed flours, total phenolic content, antioxidant activity, antibacterial activity, mineral content

## Abstract

Recently, there is an increasing demand for functional yoghurts by consumer, especially those produced through the incorporation of food of plant origin or its bioactive components. The current research was devoted to formulating functional buffalo yoghurt through the addition of 0.1 and 0.2% of fenugreek (*Trigonella foenum-graecum*) seed flour (F1 and F2) and *Moringa oleifera* seed flour (M1 and M2). The effects of fortification were evaluated on physicochemical, total phenolic content (TPC), antioxidant activity (AOA), the viability of yoghurt starter, and sensory acceptability of yoghurts during cold storage. *Moringa oleifera* seed flour had higher contents of TPC (140.12 mg GAE/g) and AOA (31.30%) as compared to fenugreek seed flour (47.4 mg GAE/g and 19.1%, respectively). Values of TPC and AOA significantly increased in fortified yoghurts, and M2 treatment had the highest values of TPC (31.61, 27.29, and 25.69 mg GAE/g) and AOA (89.32, 83.5, and 80.35%) at 1, 7, and 14 days of storage, respectively. M2 showed significantly higher antibacterial activity against *E. coli*, *S. aureus*, *L. monocytogenes*, and *Salmonella* spp. and the zones of inhibition were 12.65, 13.14, 17.23 and 14.49 mm, respectively. On the other hand, control yoghurt showed the lowest antibacterial activity and the zones of inhibition were (4.12, 5.21, 8.55, and 8.39 mm against *E. coli*, *S. aureus*, *L. monocytogenes*, and *Salmonella* spp., respectively). Incorporation of 0.1% and 0.2% of moringa seed flour (M1 and M2) led to a higher content of Ca, P, K, and Fe and lower content of Mg and Zn as compared to F1 and F2, respectively. Thus, it could be concluded that fenugreek and *Moringa oleifera* seed flour can be exploited in the preparation of functional novel yoghurt.

## 1. Introduction

Worldwide, yoghurt is considered one of the most popular fermented dairy products due to not only for its nutritional value but also for its health benefits [1]. Buffalo milk is much preferred by consumers for its rich nutrition and is drunk or transformed into valuable products such as cheese, curd, yogurt, and ice cream. Buffalo milk contains about twice as much butterfat as cow milk and higher amounts of total solids and casein, making it highly suitable for processing various types of yogurt and resulting in creamy textures and rich flavor profiles. Although its many healthy and nutritious impacts are well-established, milk and its products are generally not regarded as a rich source for particular bioactive ingredients such as polyphenols and antioxidants [2]. Thus, the formulation of novel dairy products using medicinal herbs or their extracts has gotten more attention to meet the demand of health-conscious consumers [3]. In this context, several new fermented dairy products enhanced with plant-derived foods (fruit, vegetables, or even their by-products) have been created and assessed [4]. Fenugreek (*Trigonella foenum-graecum*) is an annual plant indigenous to India and North Africa which has a lengthy background of using a range of circumstances, including diabetes and hyperlipidemia, as traditional herbal medicine. Fenugreek seeds and leaves are used in food as well as in medicinal applications, which is an old practice of human history [5]. It is widely known for its high fiber, gum, and other phytochemical components. Fenugreek seed dietary fiber, forms about 25%, has beneficial effects on digestion and can also actually change the texture of the food. In addition, polyphenol compounds such as rhaponticin and isovitexin [6], flavonoids, alkaloids, amino acids, coumarins, vitamins, saponins, and other antioxidants are thought to be the main bioactive elements in fenugreek seeds [7]. Fenugreek seeds are also a rich source for vitamins, minerals, and antioxidants. Other fenugreek components include carbohydrates, principally mucilaginous fiber (galactomannans), fixed oils (lipids), volatile oils, free amino acids, calcium, and iron, …., etc. [8]. Antidiabetic, antioxidant, anticarcinogenic, hypoglycemic activity, hypocholesterolemic activity are the major medicinal properties of the fenugreek demonstrated in various studies. Based on these several healthful benefits, fenugreek can be recommended and be a part of our daily diet and incorporated into foods to produce functional foods [9]. Thus, methi, ground fenugreek seed drink, was used in ancient Egypt to ease birth and increase milk flow, and is still used by modern Egyptian women today to ease menstrual cramps and in making hilba tea out of it to ease other types of abdominal pain. *Moringa* (*Moringa oleifera*), a Moringaceae species drumstick plant, has several medicinal advantages including injury healing, antitumor, hypotensive, anti-hepatotoxic, anti-inflammatory, antiulcer, hypocholesterolaemic, antibacterial and anti-diabetes activities. *Moringa oleifera* has large amounts of vitamin A, proteins, carbohydrates, minerals. Therefore; it is commonly used to improve nutritional status. *Moringa oleifera* has multifunctional use and has vital nutritional, industrial and medicinal applications [10]. Moreover, *Moringa oleifera* Lam. is a fast-growing tree with interesting benefits for human health [11] Nutritionally, it is possible to combine all parts of *Moringa oleifera* (leaves, seeds, fruits, immature pods, and flowers) with traditional food for human consumption [12]. In this sense, *Moringa oleifera* or its extracts have been used to improve the nutritional value of yoghurt and cottage cheese [13], with special reference to protein, fiber, and minerals [14]. *Moringa oleifera* is an abundant source of polyphenols, flavonoids, minerals, alkaloids and proteins. Substances such as bioactive carotenoids, tocopherols and vitamin C showed health-promoting potential in maintaining a balanced diet and protecting against free-radical damage that might initiate many diseases [15].

Unlike moringa, which is deeply studied and incorporated in several forms to yoghurt formulations, fenugreek has not yet been incorporated in yoghurt formulations. Consequently, the main objective of this study was to develop a functional yoghurt fortified with fenugreek seed flour (0.1 and 0.2%) and, for comparative purposes, yoghurt formulations fortified with *Moringa oleifera* seed flour (0.1 and 0.2%). Then, the effect of fortification was assessed in both formulated yoghurts by exploring the physicochemical characteristics, the viability of starter culture, as well as the mineral content and antioxidant activity, during cold storage. Moreover, the antibacterial effect of yoghurt supernatant was conducted against some pathogenic bacteria including *E. coli*, *S. aureus*, *L. monocytogenes and Salmonella* spp.

## 2. Materials and Methods

### 2.1. Chemicals, Reagents, and Culture

Buffalo milk was obtained from the herd of the Faculty of Agriculture, Cairo University, Cairo, Egypt. DVS ThermophilicYoFlex® starter culture consisting of *Streptococcus thermophiles* and *Lactobacillus delbrueckii subsp. bulgaricus* (Chr. Hansen, Horsholm, Denmark) was used in yoghurt manufacturing. MRS agar and M17 agar, the media of agar plates used for pathogenic bacteria, Salmonella Shigella agar for *S. Typhimurium*, mannitol salt agar for *S. aureus*, MacConkey sorbitol agar for *E. coli*, and Oxford agar for *L. monocytogenes*, were obtained from Biolife Italiana (Milano, Italy).

All solvents used for extraction and analyses through this study were of analytical grade. The reagent 2, 2-diphenyl-1-picrylhydrazyl (DPPH), Folin-Ciocalteu, and gallic acid were obtained from Sigma-Aldrich (*Sigma*-Aldrich, Darmstad, Germany).

### 2.2. Seed Flours Preparation

Fenugreek and *Moringa oleifera* seeds were purchased from a local market at Giza, Egypt. They were soaked in water for 15 min to remove impurities. The seeds were dehydrated in an air drier at 55 °C and ground to obtain their flours which passed through a 60-mesh sieve to obtain a uniform material. The flour was packed in polyethylene bags, and frozen (−18 °C) and used in yoghurt processing during 30 days.

### 2.3. Yoghurt Processing

Buffalo’s milk contains 6.1% fat, 3.9% protein, and 14.9% total solids (TS) were used for yoghurt processing according to [16]. Fenugreek and *Moringa oleifera* seed flour were incorporated at 0.1–0.5% to buffalo milk before heat treatment for 15 min as rehydration time with stirring. Then, it heat treated until reaching 90 °C for 5 min, cooled to 42 °C, inoculated with DVS starter culture (2%), and incubated at 42 °C ± 1 °C. After achieving a pH of 4.6 (~2.5–3 h), yoghurt samples were stored at 5 °C for 14 days. Sampling points were carried out at 1, 7 and 14 days of cold storage and subjected to chemical, microbiological and organoleptic analyses at a regular interval of 7 days as well as the determination of total phenolic content (TPC), antioxidant activity (AOA %). The antibacterial activity assay was evaluated using fresh yoghurt supernatant (Yoghurt samples were centrifuged at 4 °C for 30 min at 4000 rpm (centrifuge model C-28 AC BOECO, Hamburg, Germany) and the supernatant was filtered through a 0.45-µm Millipore membrane filter. The supernatant filtrates were kept at −20 °C until analysis [17] while mineral content was estimated at 1 and 14 days of cold storage. All analyses were carried out in triplicate.

### 2.4. Preliminary Study

After the manufacturing of the different formulations of yoghurt, a preliminary sensory evaluation study was conducted to select the best formulations that will be used throughout the experiment. A preliminary sensory evaluation study conducted on these yoghurt formulations showed that only yoghurt formulations containing 0.1% and 0.2% of fenugreek and *Moringa oleifera* seed flours were acceptable and observed that no precipitation found in the bottom of the cup after the period of coagulation which means that the added amounts have been dissolved. Thus, only these formulations were subjected to different analyses in the current research.

### 2.5. Analytical Methods of Seed Flours

#### 2.5.1. Proximate Analysis

The proximate analysis of seed flours was determined according to the methods described in AOAC [18]. Moisture, ash, and crude fiber contents were determined by the gravimetric method (AOAC 934.01), dry incineration in a muffle furnace (AOAC 942.05), and soxhlet method (AOAC 954.02), respectively. Protein content was determined by the Kjeldahl method (AOAC 976.05) and the obtained results expressed the total nitrogen content that was multiplied with factor 6.25 to obtain the total protein content. Total carbohydrate (TC) was calculated by difference (TC%) = 100 − (moisture + protein + fat + ash).

#### 2.5.2. Individual Polyphenols

The individual polyphenols of fenugreek and moringa seed flour were identified using the HPLC technique according to [19]. These compounds have been identified using HPLC by comparison with the retention times of the mix standards of Gallic acid, Catechin, Gentisic acid, Protocatechuic acid, Vanillic acid, Caffeic acid, Syringic acid, Chlorogenic, and p-Coumaric acid. We extracted 5 g of fenugreek and moringa seed flour with methanol (50 mL) and centrifuged it for 10 min at 1000 rpm (centrifuge model C-28 AC BOECO, Hamburg, Germany). The supernatant was filtered through a 0.2 μm Millipore membrane filter. 20 μL of the filtrate was injected into HPLC on Gemini-Nx 5u, C18, 250 × 4.6 mm column operated at 30 °C. Analyses were performed on the liquid chromatography HPLC Knauer, Germany, UV detector at 284 nm. The separation is achieved using a ternary linear elution gradient with (A) HPLC grade water 0.2% H3PO4 (*v*/*v*) (96%), (B) methanol (2%) and (C) acetonitrile (2%). The injected volume was 20 μL.

#### 2.5.3. TPC and AOA

For the determination of TPC and AOA, 100 mL of an aqueous methanolic solution (75%) was poured into a beaker containing 10 g of seeds flour. The beaker was covered using aluminum foil with allow stirring for 30 min. Then, it was filtered using a Whatman No. 1 filter paper. The filtrate was used in the determination of TPC and AOA. TPC was determined by using Folin-Ciocalteau reagent according to the method described by [20] and the results expressed as mg GAE/g of the sample using Gallic aid as a reference standard. While AOA% was evaluated by using DPPH according to [21].

### 2.6. Analytical Methods of Milk and Yoghurt Samples

#### 2.6.1. Physicochemical Analysis

All chemical analyses of buffalo milk and yoghurt samples were carried out in triplicate. TS, fat and protein contents of milk used for the yoghurt production were determined according to [22]. The pH values of yoghurt samples were measured by pH meter (Hanna, digital pH meter, Barcelona, Spain) while titratable acidity (as lactic acid %) was determined according to [19]. Syneresis was estimated according to [23].

#### 2.6.2. Susceptibility to Syneresis (STS)

STS of yoghurt samples was determined according to the method described by [23]. The following formula was used to calculate STS:STS = (V1/V2) × 100
where: V1 = Volume of whey collected after drainage; V2 = Volume of yoghurt sample.

#### 2.6.3. Phenolic Extraction and TPC and AOA Estimation

Control yoghurt and fortified yoghurt samples were extracted for obtaining the phenolic compounds according to the method described by [24]. In brief, ten grams of yoghurt samples were extracted with 100 mL of aqueous methanolic solution (75%) in a beaker wrapped with aluminum foil with allow stirring for 15 min with vortex-mixer (VELP Scientific, Usmate Velate, Italy). Then, the mixture was centrifuged at 4 °C for 10 min at 7200 rpm. The obtained supernatants were filtered by Whatman No.1 and the collected methanolic extracts were used in TPC and AOA assay as follows:

##### TPC of Yoghurt

TPC of yoghurt samples was determined using the micro-scale Folin–Ciocalteau method described by [25]. TPC values were expressed as mg GAE/g based on a gallic acid standard curve.

##### AOA of Yoghurt

Antioxidant activity was carried out according to [26] with slight modification. In brief, an aliquot (1.5 mL) of the obtained methanolic extract (of the yoghurt sample) was mixed with 1.5 mL of DPPH solution and was kept in the dark for 30 min at ambient temperature. The absorbance of the solution was then measured by a spectrophotometer at 517 nm. The percentage decrease in absorbance of the sample relative to the control was calculated as the relative scavenging activity [27].

#### 2.6.4. Yoghurt’s Minerals Content

P, Ca, K, Fe, Zn, and Mg were determined by inductively coupled plasma–atomic emission spectrometry (ICP-OES) using iCAP 6000 Series (Thermo Scientific, New York, NY, USA) according to [28] as follow: Samples (0.5 g) were digested with 7 mL of HNO3 (65%) and 1 mL of H2O2 (30%) (Sigma-Aldrich, St Louis, MO, USA) in ETHOS1 advanced Microwave Digestion system (Milestone, USA) for 31 min and diluted to 100 mL with deionized water [29]. Blank digestion was carried out in the same way (digestion conditions for microwave system were: 2 min for 250 W, 2 min for 0 W, 6 min for 250W, 5 min for 400 W, 8 min for 550 W, vent: 8 min, respectively). Analysis of trace elements in yoghurt samples was performed by inductively coupled plasma-atomic emission spectrometry using iCAP 6000 Series (Thermo Scientific, New York, NY, USA). All samples were analyzed in triplicates by ICP-OES. The operational parameters were as follow: RF applied power 1150 (W), Argon external flow rate 12 (L min^−1^), Argon intermediate flow rate 0.50 (L min^−1^), Argon nebulizer flow rate 0.70 (L min^−1^), Integration time 5 (s), sample uptake delay 30 (s), stabilization time 5 (s) and sample uptake rate 1.0 (mL min^−1^).

#### 2.6.5. Microbiological Analysis

Viable counts of *S. thermophilus* and *L. delbrueckii subsp. bulgaricus* in yoghurt samples at different sampling points were determined using the standard plate count method according to [30]. M17 and MRS agar culture media were used for the enumeration of *S. thermophilus* and *L. bulgaricus*, respectively. The plates were incubated in anaerobic conditions at 42 °C for 48 h or 37 °C for 72 h for the enumeration of *S. thermophilus* and *L. bulgaricus*, respectively. The results were expressed as log number of colony-forming units per g (cfu/g).

#### 2.6.6. Antibacterial Activity of Yoghurt Supernatant

For studying the antibacterial activity of yoghurt supernatant, yoghurt samples were subjected to centrifugation at 4000 rpm for 30 min at 4 °C (centrifuge model C-28 AC BOECO, Hamburg, Germany). The obtained supernatants were filtered using Millipore membrane filter of 0.45-µm diameter. The supernatant filtrates were used in determining the antibacterial activity against some pathogens (*E. coli*, *S. aureus*, *L. monocytogenes* and *S. Typhimurium*). The media of agar plates used in this study were Salmonella shigella agar for *S. typhimurium*, mannitol salt agar for *S. aureus*, MacConkey sorbitol agar for *E. coli*, and Oxford agar for *L. monocytogenes* [31]. A 100-μL diluted yogurt sample was spread on agar plates. After incubation at 37 °C for 2 days, the cell colonies were counted. Antimicrobial activity was evaluated by measuring the zones of inhibition against the tested bacteria (mm). Each assay was carried out in triplicate.

#### 2.6.7. Sensory Evaluation

Twelve trained panelists belonging to the staff member of Dairy Science Dept., Faculty of Agriculture, Cairo University were recruited for assessing the sensory descriptors of plain and fortified-yoghurts on days 1, 14 of cold storage [32]. All these trained panelists were food technology experts and were chosen based on their desire to participate and their knowledge about dairy products. They were frequent yoghurt consumers and did not have any allergies to it. Moreover, the panelists were subjected to two training sessions to study and discuss the various tested sensory descriptors including the changes in color, texture, and flavor of yoghurt samples. The panelists were instructed to wash their mouths with low sodium spring water (Safi, Egypt) during the sensory evaluation session and they were encouraged to write down any criticisms on the tested products.

Plain- and fortified-yoghurt samples were presented in plastic coded (three-digit random codes) cups. Each cup contains 50 mL of yoghurt samples that freshly removed from the refrigerator. The panelists were asked to evaluate the samples using a five-point scale where 1 = I do not like it at all, 5 = I like it extremely. The sensory evaluation of the different descriptors relied on the pre-selected descriptors: color and appearance (wheying-off, white color, reddish color), mouthfeel (ropy, uniform coagulum), body and texture (absence of curd homogeneity, lumps, bubbles), taste and flavor (sweetness, acidity, bitterness), and overall acceptability (the sum of all the character’s results). The sensory evaluation was conducted using a comparative test and fresh yoghurt as a reference sample. The data were collected in specifically designed ballots.

### 2.7. Statistical Analysis

The obtained data were statistically analyzed with two-way ANOVA to identify the significant differences between the means of samples and storage period. All data were expressed as a mean ± standard deviation of three replicates. The means of results were compared by the Tukey test with a confidence interval set at 95%.

## 3. Results and Discussion

### 3.1. Composition, TPC and AOA of Seed Flours

The data are shown in Table 1 display the nutritional composition, total phenolic content (TPC, mg GAE/g) and antioxidant activity (AOA %) of fenugreek and *Moringa oleifera* seed flours. These data indicated that fenugreek seed flour had a higher content of moisture (5.30%) and crude fiber (5.88%) while *Moringa oleifera* seed flour had a higher protein content (33.37%), oil (42.56%), ash (4.33), TPC (140.12 mg GAE/g) and antioxidant activity (31.30%). Similar data previously reported by [33] demonstrated that *Moringa oleifera* seed has higher protein content (ranging between 27–33%) and is a good source of phytochemicals. Seeds of *Moringa oleifera* could be employed in dairy products since the seed incorporation resulted in limited color changes in the fortified products [34], especially when a lower amount of *Moringa oleifera* seed flour was incorporated. Our findings are in general agreement with those found by [35] who reported slightly higher amounts of protein, fat, ash and fiber of moringa seed powder. The later authors reported lower TPC and slightly higher antioxidant activity of moringa seed powder. The current results showed that total carbohydrates were 56.2% and 17.29% for fenugreek and Moringa seed flours, respectively. These data are in general accordance with [36,37]. The variations in the present study outcomes could be attributed to polyphenolic compound extraction methods, solvent degree polarity, plant geographic locations, and plant species. Unlike Moringa, Fenugreek is less studied especially in terms of its addition to dairy products. Generally, the nutritional composition (moisture, protein, fat, ash, and fiber) of fenugreek seed flour is in general agreement with that reported by [38] which indicated slightly higher fat content in Fenugreek seed. [39] reported a higher TPC content of fenugreek seeds (85.88 mg GAE/g) as compared to our obtained data (47.40 mg GAE/g). Thus, it has a higher antioxidant activity due to its higher content of polyphenolic compounds [40].

### 3.2. Individual Phenolic Compounds of Seed Flours

Phenolic compounds of seed flours obtained from fenugreek and *Moringa oleifera* were quantified by HPLC and are shown in Table 2. It was observed that different plants have various phenolic compounds and these compounds are associated with their antioxidant activity. The current results revealed that gallic acid, catechin, and protocatechuic acid are the dominant compounds detected in both seed flour. In moringa seed flour, gallic acid showed the highest concentration (17.34 mg/100 g) followed by epicatechin, caffeic acid, p-coumaric acid, catechin, and protocatechuic acid. However, gentisic acid, vanillic acid, syringic acid and chlorogenic were not detected. On the other hand, in fenugreek seed flour, vanillic acid showed the highest concentration (57.33 mg/100 g) followed by gentisic acid, protocatechuic acid, gallic acid, chlorogenic, catechin and syringic acid. However, caffeic acid, p-coumaric acid and epicatechin were not detected.

### 3.3. Changes in Yoghurt’s pH and Titratable Acidity

As shown in Table 3, the data revealed that the values of acidity gradually increased significantly while the values of pH gradually decreased significantly as cold storage progressed. These changes were considerable at seven days of cold storage and correlated with the progress in cold storage time as well as the added amount of fenugreek and moringa seed flours. Changes in acidity and pH values are well-known to be associated with the growth of yoghurt starter culture and other lactic acid bacteria and their ability to break down carbohydrate substances and organic acids formation [41]. Ref. [42] indicated that yoghurt culture is active even at low temperatures and can ferment lactose into lactic acid, resulting in pH reduction and acidity formation. Furthermore, the findings revealed that the incorporation of fenugreek seed flour resulted in higher pH values compared to plain yoghurt and moringa-containing yoghurt over 14 days during cold storage. This could be linked to the enhanced buffering capacity that occurred by the high protein content of fenugreek seed flour [43]. The changes reported for acidity and pH are generally associated with lactic acid bacteria contained in fermented dairy products. It is therefore of great importance to retain the viability of lactic acid bacteria and to keep their viable numbers at a higher rate to produce their health-promoting activities that reflect on consumer health.

### 3.4. Yoghurt Syneresis

Syneresis is the main yoghurt problem that happens due to the weak protein network that contributes to a reduction in whey protein connection intensity, hence its separation from the body of yoghurt [44]. The data that existed in Table 3 indicated that whey separation of the tested samples increased significantly with the progress of cold storage time. The separation of whey reduced by adding seed flours. The data indicated that incorporation of fenugreek seed flour led to lower syneresis than that obtained by incorporation of moringa seed flour. This may be due in part to the soluble fiber content of fenugreek seed flour that provides the properties of texture and thickening. The addition of xanthan gum [45] or quince seed mucilage [46] has reported a similar reduction in yoghurt whey separation. Ref. [47] showed that low-solid yoghurt tends to be more synerestic than high-solid yoghurt. In this sense, yoghurt samples containing fenugreek or moringa seed flours had higher total solids that could bind the released whey and thus inhibit whey drainage [48].

### 3.5. TPC and AOA of Yoghurt

Because milk and fermented dairy products are not known to contain polyphenols, it is useful to enrich them by adding food of a plant origin. Thus, the nutritional and functional values of the resulting products will be improved. Incorporation of fenugreek and moringa seed flours increased both TPC and AOA as shown in Figure 1. Moringa seed flour-fortified yoghurt showed the highest TPC and AOA followed by fenugreek-fortified yoghurt as compared to plain yoghurt. The more seed flours incorporated, the higher TPC and AOA values obtained. The current findings showed that the values of TPC and AOA decreased with the progress of cold storage. Ref. [49] reported high radical scavenging activity of fenugreek seed flour. Ref. [50] found that *Moringa oleifera* is a great source of multiple bioactive compounds including polyphenolic antioxidant compounds. The current findings are consistent with those reported by [51], which showed that the addition of *Moringa oleifera* extract increased TPC in fortified yoghurt as compared to plain yoghurt and thus reflected as higher AOA. Interestingly, the interactions between LAB and phenolic compounds represent another factor in the formation and degradation of phenolic compounds. In this context, [52] reported that the fermentation of milk by yoghurt starter culture has produced interesting findings regarding the formation and degradation of phenolic compounds. In fact, moringa leaves are known to be rich in bioactive components including quercetin and kaempferol, among others, which display strong antioxidant properties [53]. The current data support previous findings demonstrating that food of plant origin such as fenugreek and *Moringa oleifera* seeds are rich sources of polyphenols compounds associated with its strong antioxidant activity and thus are suitable for producing bioactive yoghurt products.

### 3.6. Mineral Contents of Yoghurt

The results presented in Table 4 displayed that the incorporation of fenugreek and moringa seeds flour led to increasing mineral (Ca, P, K, Mg, Zn, and Fe) content of the various products as compared to plain yoghurt. Mineral content increased with increasing the added amount of seed flours. Incorporation of 0.1% and 0.2% of moringa seed flour (M1 and M2) led to a higher content of Ca, P, K and Fe and lower content of Mg and Zn as compared to F1 and F2, respectively. This trend was observed at 1, 7 and 14 days of cold storage. Generally, mineral content increased with the progress of cold storage time. Generally, it was observed that seeds of *Moringa oleifera* and Fenugreek [54] have elevated levels of dietary minerals. Recently, it was observed that yoghurt mousses fortified with chia seeds contain higher amounts of minerals as compared with the control yoghurt mousse [55]. The supplementation of yoghurt and dairy products with the seeds flour of fenugreek and *Moringa oleifera* increased its mineral content. However, no information exists in the literature concerning the mineral content resulted from yoghurt supplementation with food of plant origin including *Moringa oleifera* or fenugreek. As far as we know, the current research is the first study which used fenugreek seed flour in formulating a novel functional yoghurt. Because of the above-mentioned, seed flours represent a rich source of several nutrients and bioactive components and can be easily incorporated into fermented dairy products in order to improve its nutritional and functional values.

### 3.7. Viability of Yoghurt Culture

The growth and viability of probiotics and yoghurt starter culture are affected by the addition of certain compounds and this correlation has been discovered to be both species-and strain-specific. However, there has been insufficient investigation into the impact of added commercial and natural preparations on the survival and viability of bacteria [56]. In this regard, the viable counts of *S. thermophilus* and *L. bulgaricus* were determined to identify the effect of fenugreek and moringa seed flours incorporation on the viable counts of yoghurt culture. As can be seen in Figure 2, significant differences were noted in the viability of yoghurt culture in the various yoghurt samples. Unexpectedly, the obtained data displayed that the addition of moringa seed flour significantly increased the viable counts of *S. thermophilus* and *L. bulgaricus*. Moreover, fenugreek seed flour addition significantly increased the viable counts of these bacteria than control and yoghurt fortified with moringa (Figure 2). *S. thermophilus* and *L. bulgaricus* displayed higher viability at 7 days then its viability declined at final storage time (14 days) keeping higher viable counts (ranged between 5.8 to 13.3 log cfu/g for *S. thermophilus* and 4.2 to 7.9 log cfu/g for *L. bulgaricus*). Generally, [57] reported that the viable counts of *Streptococcus* in yoghurt was significantly greater than that of *Lactobacillus*. A decrease in the viable number of yoghurt bacteria during cold storage was reported by [58]. The nutrients existing in the food are among the factors that influence the viability of lactic acid bacteria. So, it is expected that using plant derivatives in yoghurt formulation led to increasing the viability of *Streptococcus* and *Lactobacillus*. This increased viability can be attributed to the polyphenols and fiber that food of a plant origin contains, among other reasons. In this sense, Fenugreek seed is well-known for its elevated fiber, gum content and other phytochemicals [6] and thus has the opportunity to improve viable counts of these bacteria as reported in the current research. This is why dietary fiber provided additional sources of carbohydrates and acts as fermentable substrate leading to lactic acid bacteria growth. Similar results previously reported by [59] demonstrated this effect of dietary fiber on the viability of lactic acid bacteria in yoghurt incorporated with pineapple dietary fiber.

Regarding the concern related with the possibility of survivability of some bacterial spores after heat treatment of yoghurt milk, it was demonstrated that the more intense heat treatment applied in yoghurt production (90–95 °C for 5 min) led to killing most vegetative microorganisms [60] while the bacterial spores survive. However, many different typologies of dairy products such as yoghurt, cheese, pasteurized milk are not suitable for the germination of the inoculated spores. This effect can be attributed to the low pH (<5) and the presence of natural microflora [61].

### 3.8. Antibacterial Activity

The results of the antibacterial activity were measured by the agar well diffusion method, and the results are expressed as an inhibition zone diameter (mm) (Table 5). Yoghurt fortified with moringa and fenugreek seed flour exhibited higher antibacterial activity compared to control yoghurt. Moreover, yoghurt fortified with moringa (M2) showed significantly higher antibacterial activity against all studied pathogenic microorganisms and the zones of inhibition were (12.65, 13.14, 17.23, and 14.49 mm) for *E. coli*, *S. aureus*, *L. monocytogenes* and *Salmonella* spp., respectively, compared to the yoghurt fortified with fenugreek seed flour and control yoghurt. These results are in close agreement with other findings obtained by [62]. Ref. [63] characterized a coagulant protein that showed both flocculating and antimicrobial effects of ~99% reduction of the bacterial population. Ref. [64] also identified a peptide derived from *Moringa oleifera* and has antibacterial activity against specific human pathogens.

### 3.9. Sensory Evaluation

The sensory assessment of yoghurt and other dairy products is the cornerstone of consumer acceptance. The sensory acceptance of yoghurt fortified with fenugreek and *Moringa oleifera* seeds flours was shown in Table 6. In general, the plain yoghurt sample (C) had the highest degree for the descriptors tested (color, mouthfeel, body and texture, taste and flavor, and overall acceptability) at 1 and 14 days of cold storage compared to the yoghurt samples fortified with fenugreek (F1 and F2) and *Moringa oleifera* (M1 and M2) seed flours. The various treatments gained lower degrees of sensory evaluation with the progress of cold storage time. Regarding the overall acceptability, the highest value was obtained for plain yoghurt samples and followed by M1 and F1. Despite the decline in all tested descriptors at the final stage of cold storage, the panelists retain high acceptability for the various yoghurt samples. This observed decline in the tested descriptors could be due to the increased acidity at the final storage time that prevents the formation of aromatic components [65]. It could also be clarified in part by the decline in yoghurt acetaldehyde concentration as cold storage progresses [66].

In the light of the accumulated information, fenugreek and *Moringa oleifera* seed flours are generally recognized as safe (GRAS) by the FDA. However, it should be taken into account that fenugreek parts are not suitable for medicinal use due to the scarcity of clinical studies. Fenugreek reacts with a wide range of drugs, thus requiring medical advice regarding consumption on an individual basis. Even though no fenugreek adverse effects on human has been reported to date, testing of fenugreek toxicity effect on liver histology in animal models is the first step to open the window for future clinical trials to investigate the safety of fenugreek for applied medical uses.

## 4. Conclusions

In view of the findings obtained by the current research, it could be concluded that seeds flour of fenugreek and *Moringa oleifera* can be incorporated for formulating novel functional yoghurt. The fiber and phytochemical nutrients present in these seed flours can improve the viability of yoghurt culture since it acts as fermentable substrates for LAB. Its incorporation led to increased TPC and AOA without decreasing its sensory acceptability. In addition, yoghurt supplemented with moringa and fenugreek showed significantly higher antibacterial activity against all studied pathogenic microorganisms compared with control yoghurt. Moreover, mineral content increased through the incorporation of these seed flours. Finally, novel fermented milk products with high nutritional and functional values can be obtained through using seed flour of fenugreek and *Moringa oleifera*.

## Figures and Tables

**Figure 1 foods-09-01157-f001:**
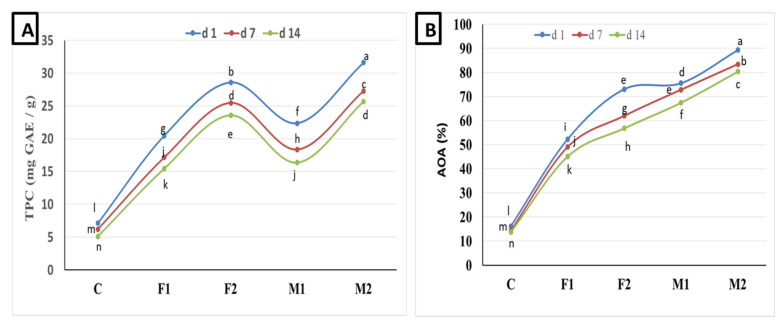
(**A**)Total phenolic content (TPC, mg GAE/g) and (**B**) antioxidant activity (AOA %) of yoghurt fortified with fenugreek and moringa seed flours during cold storage. Values are mean ± SD of three independent replicates. Lines chart with different letters are significantly different (*p* < 0.05). F1 and F2: yoghurt fortified with 0.1 and 0.2% fenugreek seed flour. M1 and M2: yoghurt fortified with 0.1 and 0.2% *Moringa oleifera* seed flour.

**Figure 2 foods-09-01157-f002:**
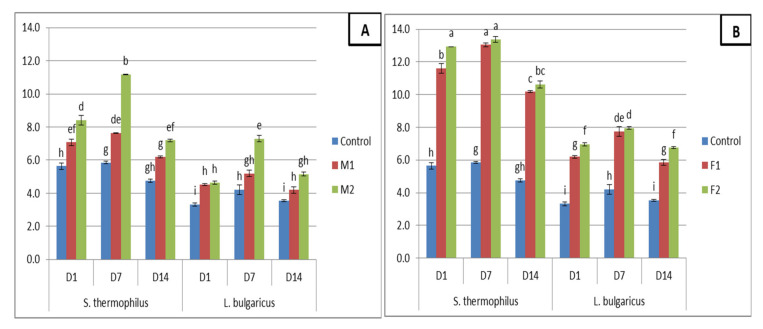
Viable counts (log CFU/g) of *S. thermophiles* and *L. delbrueckii subsp. bulgaricus* yogurt fortified with *Moringa oleifera* seeds flour (**A**) or Fenugreek seed flour (**B**). Values are mean ± SD of three independent replicates. Chart bars with different letters are significantly different (*p* < 0.05). F1 and F2: yoghurt fortified with 0.1 and 0.2% fenugreek seed flour. M1 and M2: yoghurt fortified with 0.1 and 0.2% *Moringa oleifera* seed flour.

**Table 1 foods-09-01157-t001:** Nutritional composition, total phenolic content (TPC, mg GAE/g) and antioxidant activity (AOA %) of fenugreek and *Moringa oleifera* seed flour.

Parameter	Fenugreek Seed Flour	*M. oleifera* Seed Flour
Moisture (%)	5.3 ± 0.34	2.45 ± 0.04
Protein (%)	30.7 ± 0.13	33.37 ± 0.05
Oil (%)	4.4 ± 0.26	42.56 ± 0.11
Ash (%)	3.4 ± 0.15	4.33 ± 0.04
Crude fiber (%)	7.70 ± 0.22	4.50 ± 0.2
Total carbohydrates (%)	56.2 ± 0.75	17.29 ± 0.5
TPC (mg GAE/g)	47.4 ± 0.22	140.12 ± 0.1
Antioxidant activity (%)	19.1 ± 0.66	31.3 ± 0.22

Values are mean ± SD of three independent replicates. GAE: gallic acid equivalent. Total carbohydrates (TC) = 100 − (moisture + protein + fat + ash).

**Table 2 foods-09-01157-t002:** Individual phenolic compounds (mg/100 g) of fenugreek and *Moringa oleifera* seed flours.

Compound	Fenugreek Seed	Moringa Seed Flour
Gallic acid	1.83 ± 0.02	17.34 ± 0.17
Catechin	0.54 ± 0.01	0.343 ± 0.001
Gentisic acid	36.3 ± 0.36	BDL ^1^
Protocatechuic acid	4.32 ± 0.04	0.29 ± 0.001
Vanillic acid	57.33 ± 0.57	BDL ^1^
Caffeic acid	BDL ^1^	3.21 ± 0.03
Syringic acid	0.41 ± 0.001	BDL ^1^
Chlorogenic	0.63 ± 0.01	BDL ^1^
p-Coumaric acid	BDL ^1^	0.45 ± 0.001
Epicatechin	BDL ^1^	7.74 ± 0.08

^1^ BDL: below the detection limit.

**Table 3 foods-09-01157-t003:** pH, titratable acidity (TA%) and syneresis (%) of yoghurt fortified with fenugreek and *Moringa oleifera* seed flours during cold storage.

Treatment	pH			Titratable Acidity (TA, %)			Syneresis (%)		
	d 1	d 7	d 14	d 1	d 7	d 14	d 1	d 7	d 14
C	4.59 ± 0.1 ^d^	4.40 ± 0.14 ^g^	4.10 ± 0.13 ^m^	0.86 ± 0.05 ^h^	0.95 ± 0.06 ^f^	1.06 ± 0.07 ^c^	9.82 ± 0.62 ^f^	11.30 ± 0.71 ^b^	12.39 ± 0.78 ^a^
F1	4.64 ± 0.1 ^c^	4.42 ± 0.1 ^f^	4.13 ± 0.13 ^l^	0.82 ± 0.05 ^jk^	0.84 ± 0.05 ^i^	0.85 ± 0.0 ^j^	8.74 ± 0.5 ^j^	9.60 ± 0.60 ^g^	10.85 ± 0.68 ^b^
F2	4.68 ± 0.15 ^b^	4.44 ± 0.14 ^f^	4.16 ± 0.14 ^k^	0.80 ± 0.05 ^k^	0.82 ± 0.05 ^jk^	0.83 ± 0.05 ^ij^	8.35 ± 0.52 ^k^	8.64 ± 0.54 ^j^	9.99 ± 0.63 ^e^
M1	4.58 ± 0.15 ^d^	4.33 ± 0.14 ^h^	4.04 ± 0.13 _n_	0.92 ± 0.06 ^g^	1.01 ± 0.06 ^d^	1.09 ± 0.07 ^b^	9.32 ± 0.58 ^h^	9.89 ± 0.62 ^ef^	11.33 ± 0.71 ^b^
M2	4.74 ± 0.15 ^a^	4.52 ± 0.15 ^e^	4.25 ± 0.14 ^i^	0.98 ± 0.06 ^e^	1.06 ± 0.07 ^c^	1.11 ± 0.07 ^a^	9.03 ± 0.57 ^i^	9.51 ± 0.60 ^g^	10.66 ± 0.67 ^d^

Values are means ± SD of three independent replicates. Means with different superscripts are significantly different (*p* < 0.05). C: plain yogurt, F1 and F2: yoghurt fortified with 0.1 and 0.2% fenugreek seed flour. M1 and M2: yoghurt fortified with 0.1 and 0.2% *Moringa oleifera* seed flour.

**Table 4 foods-09-01157-t004:** Minerals content (mg\kg) of yoghurt fortified with fenugreek and moringa seed flours during cold storage.

Sample	Storage Time (Days)	Minerals (mg/kg)					
		Ca	P	K	Mg	Zn	Fe
C	d 1	1426.66 ± 14.26 ^j^	1250.00 ± 12.50 ^j^	578.45 ± 5.78 ^j^	140.93 ± 1.4 ^j^	0.40 ± 0.004 ^g^	4.90 ± 0.049 ^h^
	d 14	1680.66 ± 16.80 ^h^	1310.00 ± 13.10 ^g^	647.46 ± 6.47 ^i^	187.56 ± 1.87 ^d^	0.55 ± 0.006 ^d^	5.40 ± 0.054 ^g^
M1	d 1	1760.00 ± 17.60 ^f^	1303.00 ± 13.03 ^h^	701.08 ± 7.01 ^g^	142.34 ± 1.42 ^i^	0.40 ± 0.004 ^g^	5.94 ± 0.059 ^e^
	d 14	1880.33 ± 18.80 ^c^	1375.72 ± 13.75 ^d^	784.72 ± 7.84 ^c^	189.44 ± 1.89 ^d^	0.56 ± 0.006 c	6.54 ± 0.065 ^c^
M2	d 1	1840.00 ± 18.40 ^d^	1333.30 ± 13.33 ^e^	771.19 ± 7.71 ^e^	146.57 ± 1.46 ^g^	0.42 ± 0.004 ^e^	6.53 ± 0.065 ^c^
	d 14	2046.66 ± 20.46 ^a^	1413.29 ± 14.13 ^a^	863.19 ± 8.63 ^a^	195.06 ± 1.95 ^b^	0.57 ± 0.006 ^b^	7.20 ± 0.072 ^a^
F1	d 1	1640.00 ± 16.40 ^i^	1300.00 ± 13.0 ^i^	694.14 ± 6.94 ^h^	143.75 ± 1.44 ^h^	0.41 ± 0.004 ^f^	5.88 ± 0.059 ^f^
	d 14	1768.66 ± 17.68 ^e^	1372.00 ± 13.70 ^c^	776.95 ± 7.76 ^d^	191.31 ± 1.91 ^c^	0.56 ± 0.006 ^c^	6.48 ± 0.065 ^d^
F2	d 1	1753.33 ± 17.53 ^g^	1330.00 ± 13.30 ^f^	763.55 ± 7.63 ^f^	147.98 ± 1.48 ^f^	0.42 ± 0.004 ^e^	6.47 ± 0.065 ^d^
	d14	1933.33 ± 19.33 ^b^	1409.20 ± 14.09 ^b^	854.65 ± 8.54 ^b^	196.938 ± 1.97 ^a^	0.58 ± 0.006 ^a^	7.13 ± 0.071 ^b^

Values are means ± SD of three independent replicates. Means with different superscripts are significantly different (*p* < 0.05). C: plain yogurt, F1 and F2: yoghurt fortified with 0.1 and 0.2% fenugreek seed flour. M1 and M2: yoghurt fortified with 0.1 and 0.2% *Moringa oleifera* seed flour.

**Table 5 foods-09-01157-t005:** Antibacterial activity of yoghurts fortified with fenugreek and moringa seed flours during cold storage.

Treatment	Inhibition Zone Diameter (mm)			
	*E. coli*	*S. aureus*	*L. monocytogenes*	*S. typhimurium*
C	4.12 ± 0.34 ^e^	5.21 ± 0.54 ^e^	8.55 ± 0.53 ^de^	8.39 ± 0.62 ^de^
F1	5.21 ± 0.43 ^e^	6.26 ± 0.42 ^e^	13.45 ± 0.52 ^c^	10.15 ± 0.63 ^d^
F2	6.24 ± 0.46 ^e^	7.77 ± 0.65 ^e^	14.22 ± 0.93 ^c^	12.23 ± 0.63 ^cd^
M1	9.33 ± 0.52 ^d^	10.33 ± 0.35 ^d^	21.34 ± 0.54 ^a^	18.45 ± 0.52 ^ab^
M2	12.65 ± 0.54 ^cd^	13.14 ± 0.54 ^c^	17.23 ± 0.63 ^b^	14.49 ± 0.33 ^c^

Values are means ± SD of three independent replicates. Means with different superscripts are significantly different (*p* < 0.05). C: plain yogurt, F1 and F2: yoghurt fortified with 0.1 and 0.2% fenugreek seed flour. M1 and M2: yoghurt fortified with 0.1 and 0.2% *Moringa oleifera* seed flour.

**Table 6 foods-09-01157-t006:** Sensory acceptability of yoghurt fortified with fenugreek and *Moringa oleifera* seed flours during cold storage.

Treatment	Storage Time (days)	Color	Mouthfeel	Body and Texture	Taste and Flavor	Overall Acceptability
C	d1	4.8 ± 0.10 ^a^	4.62 ± 0.10 ^a^	4.72 ± 0.10 ^a^	4.72 ± 0.10 ^a^	4.7 ± 0.1 ^a^
	d14	4.7 ± 0.10 ^b^	4.62 ± 0.10 ^a^	4.62 ± 0.10 ^b^	4.52 ± 0.09 ^a^	4.6 ± 0.1 ^b^
F1	d1	4.3 ± 0.09 ^c^	4.43 ± 0.09 ^b^	4.52 ± 0.09 ^b^	4.23 ± 0.09 ^c^	4.4 ± 0.1 ^c^
	d14	4.0 ± 0.08 ^d^	4.23 ± 0.09 ^c^	4.43 ± 0.09 ^b^	4.13 ± 0.09 ^c^	4.2 ± 0.09 ^d^
F2	d1	4.1 ± 0.09 ^d^	4.13 ± 0.09 ^d^	4.62 ± 0.10 ^b^	4.13 ± 0.09 ^d^	4.3 ± 0.09 ^d^
	d14	3.9±0.08 ^d^	4.03±0.08 ^e^	4.52±0.09 ^c^	4.33±0.09 ^d^	4.2±0.09 ^e^
M1	d1	4.5 ± 0.09 ^f^	4.33 ± 0.09 ^e^	4.33 ± 0.09 ^c^	4.23 ± 0.09 ^e^	4.4 ± 0.1 ^f^
	d14	4.1 ± 0.09 ^f^	4.03 ± 0.08 ^f^	4.62 ± 0.10 ^d^	4.03 ± 0.08 ^e^	4.2 ± 0.1 ^f^
M2	d1	4.2 ± 0.09 ^f^	4.13 ± 0.09 ^f^	4.62 ± 0.10 ^d^	4.33 ± 0.09 ^e^	4.3 ± 0.1 ^f^
	d14	4.1 ± 0.09 ^f^	4.03 ± 0.08 ^f^	4.43 ± 0.09 ^e^	4.13 ± 0.09 ^f^	4.2 ± 0.09 ^f^

Values are means ± SD of three independent replicates. Means with different superscripts are significantly different (*p* < 0.05). C: plain yoghurt, F1 and F2: yoghurt fortified with 0.1 and 0.2% fenugreek seed flour, M1 and M2: yoghurt fortified with 0.1 and 0.2% *Moringa oleifera* seed flour.

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
