# Peer review of "Antioxidant, Antibacterial Activities and Mineral Content of Buffalo Yoghurt Fortified with Fenugreek and Moringa oleifera Seed Flours"

_foods, 2020, doi:10.3390/foods9091157_

Round 1

Reviewer 1 Report

Abstract

Line 16: Correct as follows “especially THOSE produced through the incorporation of food ….”

Line 17: Correct as follows  “functional BUFFALO yogHurtS”

Line 19: Correct as follows  “The effects of fortifications were evaluated on physicochemical, total phenolic content (TPC), antioxidant activity (AOA), viability of yogurt starter, and sensory acceptability of yoghurts during cold storage”

Line 20: Correct as follows ”yogHurt starter”

Line 24: Correct as follows “…at 1, 7 and 14 days of storage”

Line 25-26: “M2 showed significant higher antibacterial activity against E. coli, S. aurous monocytogene and Salmonella spp. (12.65, 13.14, 17.23 and 14.49, respectively)”. Add the bracket after “respectively”and add the measure unit for reported results What about the antibacterial activity of other formulations?

What about the minerals?

Introduction

Line 34: “Thus, fermented foods, including dairy products, represent valuable part of the tradition diet.” Redundant period, please delete it.

Line 35: “Although its…”

Line 40: “…for more functionality values and the flavor of new  products.” Please maker clearer the period.

Line 55-64: Please expand the background on Moringa oleifera with more recent bibliography such as:

  • Fejér, J., Kron, I., Pellizzeri, V., Pľuchtová, M., Eliašová, A., Campone, L., ... & Konečná, M. (2019). First report on evaluation of basic nutritional and antioxidant properties of Moringa oleifera Lam. from Caribbean Island of Saint Lucia. Plants, 8(12), 537.

Line 65 Correct as follows “Unlike moringa which IS deeply studied…”

Line 69 Rephrase the aim of the study as follows “…. develop functional yoghurt incorporated with fenugreek seedflour (0.1 and 0.2 %) and, for comparative purposes, yoghurt formulations incorporated with Moringa oleifera seed flour (0.1 and 0.2 %). Then, the effect of fortification was assessed in both formulated yoghurts by exploring the physicochemical characteristics, the viability of starter culture, as well as the mineral content and antioxidant activity, during cold storage.”

Line 71: what do you mean of yoghurt supernatant?

Material and methods

2.5.1 Proximate analysis

It Is not enough to cite the methods you employed for the proximate analysis. You should briefly describe them

2.5.2. Individual polyphenols

You should provide more information on the HPLC method and the sample preparation (completely missing). What was the HPLC gradient? What polyphenol standards did you employ?

2.6.4. Yoghurt’s minerals content

The ICAP 6000 is an Inductively Coupled Plasma Optical Emission Spectrometry (not ICP-AES but ICP-OES).

Please add the operating conditions of the ICP-OES for the mineral analysis

Lines 157-158: “ETHOS1 advanced Microwave Digestion system (Milestone, USA) for 31 min and diluted to 100 ml with deionized water” You could specify that this microwave system has been used in different previous works, such as:

- Bua, G. D., Albergamo, A., Annuario, G., Zammuto, V., Costa, R., & Dugo, G. (2017). High-throughput icp-ms and chemometrics for exploring the major and trace element profile of the Mediterranean sepia ink. Food analytical methods, 10(5), 1181-1190.

- Albergamo, A., Mottese, A. F., Bua, G. D., Caridi, F., Sabatino, G., Barrega, L., ... & Dugo, G. (2018). Discrimination of the Sicilian prickly pear (Opuntia Ficus‐Indica L., cv. Muscaredda) according to the provenance by testing unsupervised and supervised chemometrics. Journal of food science, 83(12), 2933-2942.

Lines 160-162: “Analysis of trace elements in yoghurt samples was performed by inductively coupled plasma-atomic emission spectrometry (ICP-AES  using iCAP 6000 Series (Thermo Scientific, USA)”It is not ICP_AES, please delete the period.

Results and discussion

Lines 221-223: “These compound have been identified using HPLC by comparison with the retention times of the mix standards of Gallic acid, Catechin, Gentisic acid, Protocatechuic acid, Vanillic acid, Caffeic acid, Syringic acid, Chlorogenic, and p-Coumaric acid”. Please move this period to the section 2.5.2. Individual polyphenols

Lines 378-379: Add the bracket at the end of period and add the measure unit for reported results (12.65, 13.14, 17.23, and 14.49 for  E. coli, S. aureus, L. monocytogene and Salmonella spp., respectively)

Author Response

Reviewer 1

Comment

Response

Line 16: Correct as follows “especially THOSE produced through the incorporation of food ….”

Has been corrected line 16

Line 17: Correct as follows “functional BUFFALO yogHurtS”

Has been corrected line 17

Line 19: Correct as follows “The effects of fortifications were evaluated on physicochemical, total phenolic content (TPC), antioxidant activity (AOA), viability of yogurt starter, and sensory acceptability of yoghurts during cold storage”

Has been corrected line 19-21

Line 20: Correct as follows”yogHurt starter”

Has been corrected line 20

Line 24: Correct as follows “…at 1, 7 and 14 days of storage”

Has been corrected line 27

Line 25-26: “M2 showed significant higher antibacterial activity against E. coli, S. aurous monocytogene and Salmonella spp. (12.65, 13.14, 17.23 and 14.49, respectively)”. Add the bracket after “respectively”and add the measure unit for reported results What about the antibacterial activity of other formulations?

The brackets were added

The higher and lower values were mentioned

Lines 29-31

What about the minerals?

The data of minerals have been added line 29-31-33

Introduction

Line 34: “Thus, fermented foods, including dairy products, represent valuable part of the tradition diet.” Redundant period, please delete it.

Has been deleted line 45

Line 35: “Although its…”

Has been corrected line 46

Line 40: “…for more functionality values and the flavor of new products.” Please maker clearer the period.

The sentence modified line 50

Line 55-64: Please expand the background on Moringa oleifera with more recent bibliography such as:

  • Fejér, J., Kron, I., Pellizzeri, V., Pľuchtová, M., Eliašová, A., Campone, L., ... & Konečná, M. (2019). First report on evaluation of basic nutritional and antioxidant properties of Moringa oleifera Lam. from Caribbean Island of Saint Lucia. Plants, 8(12), 537.

The reference has been added lines 74-75

Line 65 Correct as follows “Unlike moringa which IS deeply studied…”

Has been corrected line 83

Line 69 Rephrase the aim of the study as follows “…. develop functional yoghurt incorporated with fenugreek seedflour (0.1 and 0.2 %) and, for comparative purposes, yoghurt formulations incorporated with Moringa oleifera seed flour (0.1 and 0.2 %). Then, the effect of fortification was assessed in both formulated yoghurts by exploring the physicochemical characteristics, the viability of starter culture, as well as the mineral content and antioxidant activity, during cold storage.”

Has been corrected lines 85-89

Line 71: what do you mean of yoghurt supernatant?

Preparation of yogurt supernatant

Yogurt samples were centrifuged at 4 °C for 30 min at 4000 rpm (centrifuge model C-28 AC BOECO, Germany) and the supernatant was filtered through a 0.45-µm Millipore membrane filter. The supernatant filtrates were kept at −20 °C until analyzed.

Has been explained line 122-124

Reference (17)

Material and methods

2.5.1 Proximate analysis

It Is not enough to cite the methods you employed for the proximate analysis. You should briefly describe them

The methods have been briefly described lines 139-167

2.5.2. Individual polyphenols

You should provide more information on the HPLC method and the sample preparation (completely missing).

Sample preparation has been added and more information on the HPLC method and the sample preparation has been explained

lines 172-178

What was the HPLC gradient?

Described Line 178-181

What polyphenol standards did you employ?

Described Lines 172-174

2.6.4. Yoghurt’s minerals content

The ICAP 6000 is an Inductively Coupled Plasma Optical Emission Spectrometry (not ICP-AES but ICP-OES).

Has been corrected 222

Please add the operating conditions of the ICP-OES for the mineral analysis

The operational parameters were added lines 227-232

Lines 157-158: “ETHOS1 advanced Microwave Digestion system (Milestone, USA) for 31 min and diluted to 100 ml with deionized water” You could specify that this microwave system has been used in different previous works, such as:

- Bua, G. D., Albergamo, A., Annuario, G., Zammuto, V., Costa, R., & Dugo, G. (2017). High-throughput icp-ms and chemometrics for exploring the major and trace element profile of the Mediterranean sepia ink. Food analytical methods, 10(5), 1181-1190.

- Albergamo, A., Mottese, A. F., Bua, G. D., Caridi, F., Sabatino, G., Barrega, L., ... & Dugo, G. (2018). Discrimination of the Sicilian prickly pear (Opuntia Ficus‐Indica L., cv. Muscaredda) according to the provenance by testing unsupervised and supervised chemometrics. Journal of food science, 83(12), 2933-2942.

Were explained in lines 222-224

Lines 160-162: “Analysis of trace elements in yoghurt samples was performed by inductively coupled plasma-atomic emission spectrometry (ICP-AES  using iCAP 6000 Series (Thermo Scientific, USA)”It is not ICP_AES, please delete the period.

The period has been deleted line 221

Results and discussion

Lines 221-223: “These compound have been identified using HPLC by comparison with the retention times of the mix standards of Gallic acid, Catechin, Gentisic acid, Protocatechuic acid, Vanillic acid, Caffeic acid, Syringic acid, Chlorogenic, and p-Coumaric acid”. Please move this period to the section 2.5.2. Individual polyphenols

Has been moved 172-174

Lines 378-379: Add the bracket at the end of period and add the measure unit for reported results (12.65, 13.14, 17.23, and 14.49 for  E. coli, S. aureus, L. monocytogene and Salmonella spp., respectively)

The brackets and the unit have been added  been  lines 463 -464

Reviewer 2 Report

Fortification of dairy products with plant derived ingredients is topical given consumers interest in the notion of health enhancing food formulations. The authors attempted to compare changes to compositional and biological properties of yogurt fortified containing added Fenugreek seed powder with that using added Moringa – a more established research history appears to be available in the case of the latter.

At a superficial level, the study appears to have been well executed. However, there are a some basic scientific principles that appear to have been ignored. Firstly, no consideration is given to the solubility of both plant-based seed flours. Plant-derived flours generally have impaired solubility in water (and by inference in yogurt milk, also), however, it is appropriate to characterise this property in order to see what proportion of the powder/protein is functional and what (insoluble proportion) will be inertly dispersed throughout the yogurt matrix. This is all the more important given that a comparison is being made between flours derived from 2 plant sources. Furthermore, the authors make an overt reference to the prebiotic potential of whatever soluble fibre is present in these seed flours. Presumably, soluble fibre is only accessible following powder solubilisation. Equally, all these background factors are likely to impact on the measured (aqueous-derived) antioxidant values while the measured total phenolic content (TPC) values would have been derived using more potent organic chemical extractants. The apparent increase in the measured mineral values of all prepared yogurts during storage merits comment. The fact that this also occurs in the case of the Control raises a question as to whether changing pH values affects the analytical methodology!

From a sensory viewpoint, yogurt consistency and whey-off is affected considerably by milk preheat treatment and associated induced interactions between K-casein and B-lactoglobulin. While the syneresis results look better for the seed-flour fortified yogurt, it would have been appropriate to provide some background characterisation of protein denaturation, particularly since both flours contain over 30% protein.

A question mark exists over the microbiological aspect of the study. The plant derived flours were purchased on the open market without any microbiological specification. It is highly like that such flours could contain microbial spores as a result of environmental contamination and handling. Such spores would survive preheat treatment and may subsequently germinate and grow during yogurt storage, resulting in spoilage and/or altered fermentation.

The Introduction is a superficial review of putative biofunctional properties. A more in depth scrutiny of the available literature is warranted in order to anticipate formulation challenges and their impacts on subsequent functionality e.g. antioxidant activity.

The authors should review the compositional data shown in Table 1 as the fat (%) appears very low at 4.4%. Either this constituent or another needs to be upgraded so as to reach an aggregate compositional value of 100%.

In Materials and Methods, the method of the dispersal of the seed flour powders in yogurt milk should be described including any rehydration time allowed before moving to the next processing step (e.g. preheat treatment)

Finally, the authors should have regard to concerns and risks about consuming Fenugreek particularly in light of its interactions with other food supplements as well as prescription drugs. The acceptability, or otherwise, of the 1% and 2% fortification levels applied should be discussed in light of the aforementioned concerns.

Author Response

Comment

Response

At a superficial level, the study appears to have been well executed. However, there are some basic scientific principles that appear to have been ignored. Firstly, no consideration is given to the solubility of both plant-based seed flours. Plant-derived flours generally have impaired solubility in water (and by inference in yogurt milk, also), however, it is appropriate to characterise this property in order to see what proportion of the powder/protein is functional and what (insoluble proportion) will be inertly dispersed throughout the yogurt matrix. This is all the more important given that a comparison is being made between flours derived from 2 plant sources.

Furthermore, the authors make an overt reference to the prebiotic potential of whatever soluble fibre is present in these seed flours. Presumably, soluble fibre is only accessible following powder solubilisation. Equally, all these background factors are likely to impact on the measured (aqueous-derived) antioxidant values while the measured total phenolic content (TPC) values would have been derived using more potent organic chemical extractants. The apparent increase in the measured mineral values of all prepared yogurts during storage merits comment. The fact that this also occurs in the case of the Control raises a question as to whether changing pH values affects the analytical methodology!

The Fenugreek and Moringa oleifera seeds flour were passed through a 60-mesh sieve to obtain a fine unique material. Then incorporated to buffalo milk during heat treatment with stirring to enhance the solubility. In addition, during the preliminary study we selected this two concentration and observed that no precipitation found in the bottom of the cup after the period of coagulation which means that the added amounts have been dissolved. 

lines 110-111

lines 132-133

High protein solubility in both acidic and alkaline pH regions can be applied in food formulation; therefore fenugreek can be useful in the formulation of acid or alkaline foods as reported by Fasuyi, O. ayodeji. (2005). Varietal composition and functional properties of cassava (Manihrt esculentra cranzt). Pakistan Journal of Nutrition., 4(1): 43-49.

From a sensory viewpoint, yogurt consistency and whey-off is affected considerably by milk preheat treatment and associated induced interactions between K-casein and B-lactoglobulin. While the syneresis results look better for the seed-flour fortified yogurt, it would have been appropriate to provide some background characterisation of protein denaturation, particularly since both flours contain over 30% protein.

The results showed that Fenugreek and Moringa oleifera seeds flour supplementation of yogurt led to lower syneresis values, suggesting improved viscosity. This could be attributed to some interactions between the components of seeds flour and the proteins in the yogurt. The yogurt gel matrix seemed to increase by the addition of seeds flour, thereby being able to hold more yogurt serum.

A question mark exists over the microbiological aspect of the study. The plant derived flours were purchased on the open market without any microbiological specification. It is highly like that such flours could contain microbial spores as a result of environmental contamination and handling. Such spores would survive preheat treatment and may subsequently germinate and grow during yogurt storage, resulting in spoilage and/or altered fermentation.

The Fenugreek and Moringa oleifera seeds were soaked in water for 15 min to remove impurities. The seeds were dehydrated in air drier at 55°C.

Lines101 -103

Then, Fenugreek and Moringa oleifera seeds flour were incorporated at 0.1 - 0.5 % to buffalo milk during heat treatment at 60 ͦC with stirring. Then, the heat treatment of milk was raised to 90° C to kill any pathogenic microorganism

Lines 108-109

The Introduction is a superficial review of putative biofunctional properties. A more in depth scrutiny of the available literature is warranted in order to anticipate formulation challenges and their impacts on subsequent functionality e.g. antioxidant activity.

Lines 63-66

Antidiabetic, antioxidant, anticarcinogenic, hypoglycemic activity, hypocholesterolemic activity are the major medicinal properties of the fenugreek demonstrated in various studies. Based on these several healthful benefits, fenugreek can be recommended and be a part of our daily diet and incorporated into foods in order to produce functional foods.

Khorshidian, N., Yousefi Asli, M., Arab, M., Adeli Mirzaie, A., & Mortazavian, A. M. (2016). Fenugreek: potential applications as a functional food and nutraceutical. Nutrition and Food Sciences Research3(1), 5-16.

Lines 79-82

Moringa leaves are an abundant source of polyphenols, flavonoids , minerals , alkaloids and proteins. The substances such as bioactive carotenoids, tocopherols and vitamin C showed health-promoting potential in maintaining a balanced diet and protecting against free-radical damage that might initiate many diseases

El-Ziney, M. G., Shokery, E. S., Youssef, A. H., & Mashaly, R. E. (2017). Protective effects of green tea and moringa leave extracts and their bio-yogurts against oxidative effects of lead acetate in albino rats. J Nutrit Health Food Sci5, 1-11.

The authors should review the compositional data shown in Table 1 as the fat (%) appears very low at 4.4%. Either this constituent or another needs to be upgraded so as to reach an aggregate compositional value of 100%.

The crud fiber were upgraded to be 7.7±22.in table 1

In Materials and Methods, the method of the dispersal of the seed flour powders in yogurt milk should be described including any rehydration time allowed before moving to the next processing step (e.g. preheat treatment)

Fenugreek and Moringa oleifera seeds flour were incorporated at 0.1 - 0.5 % to buffalo milk before heat treatment for 15 min as rehydration time. Then, it heat treated until reaching 90° C for 5 min, cooled to 42 °C.

Lines 116-117

Finally, the authors should have regard to concerns and risks about consuming Fenugreek particularly in light of its interactions with other food supplements as well as prescription drugs. The acceptability, or otherwise, of the 1% and 2% fortification levels applied should be discussed in light of the aforementioned concerns.

As reported by Rao PU, Sesikeran B, Rao PS, et al. Short term nutritional and safety evaluation of fenugreek. Nutr Res 1996;16:1495-1505, it was found that feeding rats with high levels of fenugreek seeds (20 %) did not cause any  significant toxicological hematological, hepatic, or histopatho-logical changes in weanling rats fed fenugreek seeds for 90 days.

Reviewer 2

Reviewer 3 Report

Authors submit an original research which main aim is to develop a new yoghurt using buffalo milk and incorporating Fenugreek and Moringa oleifera seed flours. The nutritional functionality of the new product is based on the antioxidant and antibacterial capacity if the seed flours, and also include in the study the mineral content. According to them, it is the first paper dealing with the use of both seed flours in fermented milk to produce yoghurt. The idea is interesting based on the conceptual of new products for the dairy industry, but the nutricional functionality need to be proved further. With the current experiment and the data obtained it is too speculative to consider this new product as a Functional. The current study sounds interesting as a preliminar study that need to be more precise in future estudies. To my understanding that issue will affect to the title of the manuscript and I recommend to modify. Also some formal and substantive aspects should be clarified before further decisions. 

-Some aspects need to be justified such as the election of buffalo milk, if it is the adequate one rather than cow mils as the most common one, or sheep or goat. What makes buffalo milk the election milk.

-Also the concentration of the seed flour to 1 or 2%. Which is the reason the justify this selection. I assume that can be sensorial reasons in one case, but not sure in the other. Please, it will help to understand some effects.

-An importante issue of the study (maybe included in the title), is the evolution of the different markers for 14 days. Generally talking, the shelf life of a yoghurt is 21 days. I wonder why the study was stopped at the 2º week because some of the trends in the results could be understood at the 3erd week of shelf-life.   

-Among all the markers selected, minerals are included in this study. In my view, to study minerals in such a reducen concentration as an ingredients have not a remarkable effect on the final producto. Also the results and discussion section try to justify the meaning, but is not enough clear. Can the authors justify the election of the mineral? Also, and talking in nutritional terms, some of them can be divided into electrolytes where Na is missed. 

-Talking about lipid components of seed I will rather use oil than fat, since fat is more for solid forms. 

-It is analyzed "Crude fiber" and in several places, including the Conclusion section, author consider "soluble fiber" as one of the justifications of different parameters because is a source of carbohydrates for microbial growth or for syneresis effect justification. It is difficult to justify something when there is no information in or laboratory work. How can it be justified by the authors. 

-Authors use different abbreviations like Antioxidant capacity (AOC), but they don't use always the abbreviation in the text or in the tables. I will suggest to review the manuscript to harmonize it, and also use as footnote in the tables because all tables need to contain all the information to be understandable.  

-Also in Table 2, some of the individual phenolic compounds are indicated as ND, non detectable. I suggest to used, below detection limit, and indicates in the methodology. Regarding methodology, it is properly described in al section but in sensory evaluation, where there is not an indicated reference followed and the conditions of the environment properly described  In addition, the author mention a 5-point scale but no attribute is mentioned o described till line 389. That section need to be improved. 

-To be able to understand and review the data, it is extremely important to be clear in the statistical analyses. In all tables the letter superscript is not clear or properly analyzed. It is needed to state if the differences is within column or line. My impression is the all data within the same table has been statistically analyzed, and the comparison and justifications in the text are difficult to be understood. 

-Data in Table 2 should have an SD value.

-A general argumentar line of the author is that adding these sed flour all markers improved. My question is related to the proportion. If authors used 1 and 2%, it means that it is double amount of seed fours, but there ir not a double increase of the markers selected. How can be explained this unexpected results?

Once answered these questions and rewritten could be considered to be revised again and, in case, for publication.    

Author Response

Comment

Response

-Some aspects need to be justified such as the election of buffalo milk, if it is the adequate one rather than cow mils as the most common one, or sheep or goat. What makes buffalo milk the election milk?

Lines 40-45

Buffalo milk is much preferred by consumer for its rich nutrition and is drunk or transformed into valuable products such as cheese, curd, yogurt and ice cream. Buffalo  milk  contains  about  twice  as  much  butterfat  as  cow  milk  and higher  amounts  of  total  solids  and  casein, making  it  highly  suitable  for  processing  various types of yogurt and resulting in creamy textures and rich flavor profiles.

-Also the concentration of the seed flour to 1 or 2%. Which is the reason the justify this selection. I assume that can be sensorial reasons in one case, but not sure in the other. Please, it will help to understand some effects.

After manufacturing of the different formulations of yoghurt, a preliminary sensory evaluation study was conducted to select the best formulations that will be used throughout the experiment. Preliminary sensory evaluation study conducted on these yoghurt formulations showed that only yoghurt formulations containing 0.1 and 0.2 % of fenugreek and Moringa oleifera seed flours were acceptable. and observed that no precipitation found in the bottom of the cup after the period of coagulation which means that the added amounts have been dissolved. Thus, only these formulations were subjected to different analyses in the current research. 

Lines 128-134      

-An important issue of the study (maybe included in the title), is the evolution of the different markers for 14 days. Generally talking, the shelf life of a yoghurt is 21 days. I wonder why the study was stopped at the 2º week because some of the trends in the results could be understood at the 3erd week of shelf-life.   

We are following the Egyptian standard no.1000/1990 which says that the shelf life of yogurt with cup cover are 14 days.

-Among all the markers selected, minerals are included in this study. In my view, to study minerals in such a reducen concentration as an ingredients have not a remarkable effect on the final producto. Also the results and discussion section try to justify the meaning, but is not enough clear. Can the authors justify the election of the mineral? Also, and talking in nutritional terms, some of them can be divided into electrolytes where Na is missed. 

Moringa and fenugreek seeds have a lot of minerals that are essential for growth and development among which, calcium is considered as one of the important minerals in human nutrition. Fenugreek has a high amount of calcium, iron and zinc.

.Jani R, Udipi S, Ghugre P. Mineral content of complementary foods. Indian J Pediatr. 2009;76(1):37-44.

-Talking about lipid components of seed I will rather use oil than fat, since fat is more for solid forms. 

Has been replaced in table 1 and in line 137

-It is analyzed "Crude fiber" and in several places, including the Conclusion section, author consider "soluble fiber" as one of the justifications of different parameters because is a source of carbohydrates for microbial growth or for syneresis effect justification. It is difficult to justify something when there is no information in or laboratory work. How can it be justified by the authors. 

We are expected that because amount of fiber as a source of carbohydrate according to the data reported in literature.

Several previously reported data revealed that food-derived fiber had the ability to improve the bacterial growth including fenugreek and moringa oliveira seed flours

Rajoka, M. S. R., Shi, J., Mehwish, H. M., Zhu, J., Li, Q., Shao, D., ... & Yang, H. (2017). Interaction between diet composition and gut microbiota and its impact on gastrointestinal tract health. Food Science and Human Wellness, 6(3), 121-130.

-Authors use different abbreviations like Antioxidant capacity (AOC), but they don't use always the abbreviation in the text or in the tables. I will suggest to review the manuscript to harmonize it, and also use as footnote in the tables because all tables need to contain all the information to be understandable.  

We did not use the abbreviations of AOC along the whole manuscript, but the abbreviation AOA (antioxidant activity) have been used.

Regarding the tables footnotes, it already used under all tables

-Also in Table 2, some of the individual phenolic compounds are indicated as ND, non-detectable. I suggest to used, below detection limit, and indicates in the methodology.

ND has been replaced by BDL (below detection limit in table 2 and also the footnote

Regarding methodology, it is properly described in all section but in sensory evaluation, where there is not an indicated reference followed and the conditions of the environment properly described in addition, the author mention a 5-point scale but no attribute is mentioned or described till line 389. That section need to be improved. 

The section has been improved and the reference has been added 

Lines 254-263

Dhull, S. B., Punia, S., Sandhu, K. S., Chawla, P., Kaur, R., & Singh, A. (2020). Effect of debittered fenugreek (Trigonella foenum‐graecum L.) flour addition on physical, nutritional, antioxidant, and sensory properties of wheat flour rusk. Legume Science, 2(1), e21.

-To be able to understand and review the data, it is extremely important to be clear in the statistical analyses. In all tables the letter superscript is not clear or properly analyzed.

It is needed to state if the differences is within column or line. My impression is the all data within the same table has been statistically analyzed, and the comparison and justifications in the text are difficult to be understood. 

In all tables the letter superscripts have been clarified.

Regarding the statistical analysis, the means data have been statistically analyzed with two-way ANOVA to identify the significant differences between the means of samples and storage period.

-Data in Table 2 should have an SD value.

The SD values has been added in table 2

-A general argument line of the author is that adding these seed flour all markers improved. My question is related to the proportion. If authors used 1 and 2%, it means that it is double amount of seed fours, but there is not a double increase of the markers selected. How can be explained this unexpected results?

The effects of the seeds flour are due to some components such as phenolic compound and they’re affected by the milk components and the interaction with the components of milk has an effect that may increase or decrease the effect. Generally, not all double fortification amount can cause double increase in the studied parameters.

Reviewer 3

Round 2

Reviewer 2 Report

In my initial report, I raised a number of issues of a scientific nature about the manuscript. I would have expected that the authors would have answered these using either additional data or alternatively (in its absence) by opening up debate in the Results and Discussion section. The authors have not done either - there are few edits inserted into the Results and Discussion. Unfortunately, the authors' replies (response) attempt to brush over each  commentary e.g. a) while heating to 90 degC may kill bacteria, it will not kill spores! - this scientific fact is ignored, b) the revision of the proximate composition in Table 1 does not explain the total compositional make up of Fenugreek seed flour and c) my concerns about the solubility status of the Fenugreek seed flour is dismissed on the basis that there were residues evident in the yoghurt. This remark ignores the fact that yogurt is a viscous acid gel which can entrain and hold insoluble material in suspension.

The authors dismiss my remarks about the Health & Safety aspects of Fenugreek. It should be noted in the text e.g. the Food & Drug Administration lists Fenugreek seed as having GRAS status, but not suitable for medicinal use. Fenugreek reacts with a wide range of drugs, thus required medical advice regarding consumption on an individual basis

All these points can be explored in the Results and Discussion.

Author Response

Comments

Response and place

In my initial report, I raised a number of issues of a scientific nature about the manuscript. I would have expected that the authors would have answered these using either additional data or alternatively (in its absence) by opening up debate in the Results and Discussion section. The authors have not done either - there are few edits inserted into the Results and Discussion. Unfortunately, the authors' replies (response) attempt to brush over each commentary e.g.

a) While heating to 90 degree C may kill bacteria, it will not kill spores! - this scientific fact is ignored,

All these points can be explored in the Results and Discussion.

a)      In lines 402 – 407, the paragraph has been added  

Regarding the concern related with the possibility of survivability of some bacterial spores after heat treatment of yoghurt milk, It was demonstrated that the more intense heat treatment applied in yoghurt production (90 – 95 °C for 5 min) led to killing most vegetative microorganisms [17] while the bacterial spores survive. However, many different typologies of dairy products such as yoghurt, cheese, pasteurized milk are not suitable for the germination of the inoculated spores. This effect can be attributed to the low pH (< 5) and the presence of natural microflora.

b) the revision of the proximate composition in Table 1 does not explain the total compositional make up of Fenugreek seed flour and

b. Table 1, the proximate composition has been checked and the total carbohydrates of seed flours have been added into this Table and it has been calculated by difference.

Lines 255-257

c) My concerns about the solubility status of the Fenugreek seed flour is dismissed on the basis that there were residues evident in the yoghurt. This remark ignores the fact that yogurt is a viscous acid gel which can entrain and hold insoluble material in suspension.

C. Regarding the solubility status of seed flours in yoghurt: Based on my understanding of your question and concern about the solubility of seed flours in yoghurt, I want to refer that seed flours have been added to milk (not to yoghurt) with stirring for 15 min as rehydration time to improve its solubility. Moreover, the seed flours have been sieved using a 60-mesh sieve to obtain uniform material.

please revise line 101 – 105

The Fenugreek and Moringa oleifera seeds were soaked in water for 15 min to remove impurities. The seeds were dehydrated in air drier at 55°C and grinded to obtain their flours which passed through a 60-mesh sieve to obtain a uniform material.

Line 108– 110

Fenugreek and Moringa oleifera seeds flour were incorporated at 0.1 - 0.5 % to buffalo milk before heat treatment for 15 min as rehydration time with stirring. Then, it heats treated until reaching 90° C for 5 min, cooled to 42 °C, inoculated with DVS starter culture (2 %), and incubated at 42°C ± 1°C. After achieving a pH of 4.6 (~2.5 - 3 h), yoghurt samples were stored at 5° C for 14 days.

d) The authors dismiss my remarks about the Health & Safety aspects of Fenugreek. It should be noted in the text e.g. the Food & Drug Administration lists Fenugreek seed as having GRAS status, but not suitable for medicinal use. Fenugreek reacts with a wide range of drugs, thus required medical advice regarding consumption on an individual basis

Lines 467 – 473… paragraph about this concern has been added

In the light of the accumulated information, fenugreek and moringa oliefera seed flours are generally recognized as safe (GRAS) by FDA. However, it should be taken into account that fenugreek parts are not suitable for medicinal use due to the scarcity of clinical studies. Fenugreek reacts with a wide range of drugs, thus required medical advice regarding consumption on an individual basis. Even though no fenugreek adverse effects on human has been reported to date, testing of fenugreek toxicity effect on liver histology in animal models is the first step to open the window for future clinical trials to investigate safety of fenugreek for applied medical uses.

Reviewer 3 Report

Thank you to the authors to clarify all the comments. In general terms the manuscript has been improved and comments considered, but still there are some improvements needed. 

-About methodology, sensory evaluation section has been improved nut still need to clarify the method used to evaluate the descriptors tested: color, mouthfeel, body and texture, taste and flavor, and overall acceptability. This evaluation was made on a 5 point scale? Usually is a 9 point scale. If it was 5, which where the descriptors used to identified the different points? There was a defined end point or limit, for example 4 over a scale of 5? There the panelists trained and if ye, how? Was it a QDA or a comparative test (2 samples at the same time, 3 samples,...did the panelists used a reference samples, for example a fresh yoghurt? 

-To improve the manuscript and save space, all "proximate analysis" can be combined in one section sin they follow the same methodology, reference number 18, AOAC method. Authors can use the model of crude protein. Additionally, it is important to indicated the AOAC Official Method number, like for example 2001.11 for crude protein. Particularly for Crude Protein, the method refers to nitrogen (N) that need to be converted into protein using a facto F = factor to convert N to protein, that it is assumed that ir is 6.25 (rather than 5.70 for wheat, 6.38 for dairy products), but should be indicated.

-Authors justify the self life of yoghurt in 14 days based the Egyptian standard no.1000/1990. Looking into other references I find other references in 21 days and even 28 in some cases. To me it is still hard to believe that it is only 14 days if it is preserved at 5ºC as stated by the authors.

-Although authors made some progress on the clarification for the statistical analyses, there is still some concerns about it. Significant differences cannot be made combining different dependent variables. The differences are based on the different superscripts letter, by comparing row (storage period), or column (by different samples, plain yoghurt or with the different seed flours), and algo within the same analytical indicator, but not all at the same time. In addition the criteria of this ANOVA changes from Tables 3 and 4 to Table 5 (minerals), and Tables 1 and 2 have no statistical analyses. Since the authors are talking about trends in some cases, it will be more visual and easy to understand to present the results on a graphic like for microbial growth (Figure 1). I will suggest to change, at least, Tables 3, 4 and 5 into lines figures (not columns figures), to display the trends. the significant differences can be marked with an asterisk where needed. This kind of messy Table with such a large list of number makes very difficult to obtain any clear conclusion. 

-The 2 questions made regarding the effect of addition of 1 and 2% of seed flour, try to understand the real effect of this fortification. On one hand, authors consider that, based on other authors, these seeds are a good source of nutrients, but if this impact of enrichment where so important, when adding 2% it should be a direct double than 1%, although authors consider that "not all double fortification amount can cause double increase in the studied parameters." I agree with authors that there is an impact on the final composition of fortified yoghurts (I will modify the title from "incorporated to fortified") but less that the authors attribute to the new products. 

For reasons above mentioned, I understand that the manuscript has potential new information but still need to be improved, and will keep ranking as "major revision".

Author Response

Comments

Response and place

-About methodology, sensory evaluation section has been improved but still need to clarify the method used to evaluate the descriptors tested: color, mouthfeel, body and texture, taste and flavor, and overall acceptability. This evaluation was made on a 5 point scale? Usually is a 9 point scale. If it was 5, which where the descriptors used to identified the different points?. There was a defined end point or limit, for example 4 over a scale of 5?. There the panelists trained and if yes, how?. Was it a QDA or a comparative test (2 samples at the same time, 3 samples,...did the panelists used a reference samples, for example a fresh yoghurt? 

Lines 219 - 236

The sensory evaluation method has been improved considering all points of the reviewer.

-To improve the manuscript and save space, all "proximate analysis" can be combined in one section since they follow the same methodology, reference number 18, AOAC method. Authors can use the model of crude protein. Additionally, it is important to indicate the AOAC Official Method number, like for example 2001.11 for crude protein. Particularly for Crude Protein, the method refers to nitrogen (N) that need to be converted into protein using a facto F = factor to convert N to protein, that it is assumed that ir is 6.25 (rather than 5.70 for wheat, 6.38 for dairy products), but should be indicated.

Lines 130 – 136

The proximate analysis has been briefly discussed and put in one paragraph. The nitrogen content of seed flours has been multiplied with a factor of 6.25.

The AOAC official method number has been added.

                The proximate analysis of seed flours was determined according to the methods described in AOAC (18). Moisture, ash, crude fiber contents were determined by gravimetric method (AOAC 934.01), dry incineration in a muffle furnace (AOAC 942.05), and soxhlet method (AOAC 954.02), respectively. Protein content was determined by the Kjeldahl method (AOAC 976.05) and the obtained results expressed the total nitrogen content that was multiplied with factor 6.25 to obtain the total protein content. Total carbohydrate (TC) was calculated by difference (TC % = 100 – (moisture + protein + fat + ash)

Authors justify the shelf life of yoghurt in 14 days based the Egyptian standard no.1000/1990. Looking into other references I find other references in 21 days and even 28 in some cases. To me it is still hard to believe that it is only 14 days if it is preserved at 5ºC as stated by the authors

I confirm that the Egyptian standard recommends that the shelf-life of the sealed yoghurt (produced at industrial scale at dairy factories) has 21 days of shelf life while the unsealed yoghurt (produced at laboratories and small scale, like our case for research purpose) has 14 days of shelf life.

Please, take a look on the following rticles about yoghurt (the shelf life is 14 days):

-          Abd El-Fattah, A., Sakr, S., El-Dieb, S., & Elkashef, H. (2018). Developing functional yogurt rich in bioactive peptides and gamma-aminobutyric acid related to cardiovascular health. LWT, 98, 390-397.

-          Sofu, A., & Ekinci, F. Y. (2007). Estimation of storage time of yogurt with artificial neural network modeling. Journal of dairy science, 90(7), 3118-3125.

-Although authors made some progress on the clarification for the statistical analyses, there is still some concerns about it. Significant differences cannot be made combining different dependent variables. The differences are based on the different superscripts letter, by comparing row (storage period), or column (by different samples, plain yoghurt or with the different seed flours), and also within the same analytical indicator, but not all at the same time. In addition the criteria of this ANOVA changes from Tables 3 and 4 to Table 5 (minerals), and Tables 1 and 2 have no statistical analyses.

Since the authors are talking about trends in some cases, it will be more visual and easy to understand to present the results on a graphic like for microbial growth (Figure 1). I will suggest to change, at least, Tables 3, 4 and 5 into lines figures (not columns figures), to display the trends. the significant differences can be marked with an asterisk where needed. This kind of messy Table with such a large list of number makes very difficult to obtain any clear conclusion. 

Regarding your suggestion about presenting the data of some table into figures (lines figures) especially Table 3, 4, and 5, We agreed with this suggestion that figures are easy to understand. But, conversion of table 3 is very difficult because it has three different parameters (pH, acidity, and syneresis). Also, Table 5 (mineral content in yoghurt samples) is very difficult to make it because there is a big variance between mineral values, Ca = 1426, Zn = 0.4, and Fe = 4.9 mg / kg. So, we change the data of Table 4 into line figure as you suggested.

Figure 1

-The 2 questions made regarding the effect of addition of 1 and 2% of seed flour, try to understand the real effect of this fortification. On one hand, authors consider that, based on other authors, these seeds are a good source of nutrients, but if this impact of enrichment where so important, when adding 2% it should be a direct double than 1%, although authors consider that "not all double fortification amount can cause double increase in the studied parameters." I agree with authors that there is an impact on the final composition of fortified yoghurts (I will modify the title from "incorporated to fortified") but less that the authors attribute to the new products. 

Line 2 in the title

Thanks, dear Reviewer for this comment and we agreed with your comment about the title modification (replacement of the word incorporated and putting the word fortified)
